# Realizing LLMs' Causal Potential Requires Science-Grounded, Novel Benchmarks

## Abstract

Recent claims of strong performance by Large Language Models (LLMs) on causal discovery tasks are undermined by a critical flaw: many evaluations rely on widely-used benchmarks that likely appear in LLMs' pretraining corpora. As a result, empirical success on these benchmarks seem to suggest that LLM-only methods, which ignore observational data, outperform classical statistical approaches on causal discovery. In this position paper, we challenge this emerging narrative by raising a fundamental question: Are LLMs truly reasoning about causal structure, and if so, how do we measure it reliably without any memorization concerns? And can they be trusted for causal discovery in real-world scientific domains? We argue that realizing the true potential of LLMs for causal analysis in scientific research demands two key shifts. First, (P.1) the development of robust evaluation protocols based on recent scientific studies that effectively guard against dataset leakage. Second, (P.2) the design of hybrid methods that combine LLM-derived world knowledge with data-driven statistical methods.

To address P.1, we motivate the research community to evaluate discovery methods on real-world, novel scientific studies, so that the results hold relevance for modern science. We provide a practical recipe for extracting causal graphs from recent scientific publications released after the training cutoff date of a given LLM. These graphs not only prevent verbatim memorization but also typically encompass a balanced mix of well-established and novel causal relationships. Compared to widely used benchmarks from `BNLearn`, where LLMs achieve near-perfect accuracy, LLMs perform significantly worse on our curated graphs, underscoring the need for statistical methods to bridge the gap. To support our second position (P.2), we show that a simple hybrid approach that uses LLM predictions as priors for the classical PC algorithm significantly improves accuracy over both LLM-only and traditional data-driven methods. These findings motivate a call to the research community: adopt science-grounded benchmarks that minimize dataset leakage, and invest in hybrid methodologies that are better suited to the nuanced demands of real-world scientific inquiry.

## 1 Introduction

Causal discovery, which is the task of learning the underlying causal graph is a foundational step in many causal inference problems. For instance, in treatment effect estimation [45, 38], the causal graph identifies appropriate adjustment variables to account for confounding. In interventional and counterfactual analysis [37, 39], it reveals the pathways through which interventions influence outcomes. Traditionally, causal discovery has been dominated by data-driven methods that infer graph structure using observational datasets. These approaches typically fall into three categories: (i) constraint-based methods that apply statistical tests [15, 50] to infer conditional independence

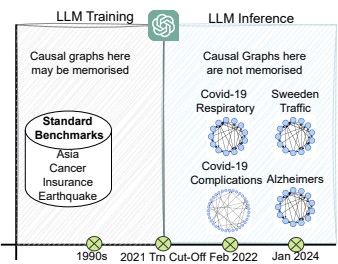

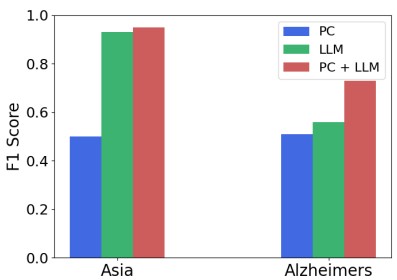

| (a) Causal Graphs Timeline | (b) Comparing Asis and Alz. Datasets |

Figure 1: **(a)** Our Novel Science benchmarks were created post-2021 and required expert consensus, unlike widely-used BNLearn graphs from the 1990s that likely appeared in LLM training data and memorized. Evaluating with LLM checkpoints trained on pre-2021 data ensures these graphs are unseen, enabling a fair benchmarking. **(b)** We compare the performance of PC, an LLM-BFS [25], and a hybrid PC+LLM approach (refer Sec. 3.1) on two graphs: Asia (pre-2021, likely included in LLM training data) and Alzheimers (post-2021, unseen during training). We observe a large gap in F1 scores between PC and LLM on the Asia dataset, a contrast that is notably diminished on the Alzheimers dataset.

relationships [49, 48], (ii) score-based methods that search for graphs optimizing a goodness-of-fit score [16, 10, 36, 60], and (iii) functional-causal models that leverage assumptions about the data-generating process, such as additive noise [17, 57] or non-Gaussian residuals [46].

Despite significant progress in causal discovery, some fundamental limitations still hinder the effectiveness of current methods [13, 14, 19, 26]. Suppose we consider a simple case with two dependent variables, $X$ and $Y$. Although we can measure their dependence using metrics such as correlation from observational data, it is impossible to identify the direction of causation. This is because both causal models: $X \rightarrow Y$ with likelihood assessed using $P(X)P(Y|X)$ and $Y \rightarrow X$ with $P(Y)P(X|Y)$ can explain the data equally well. Disambiguating between the two requires additional assumptions such as constraints on the distribution of error residuals [46], or external supervision from experts, since observational data alone cannot identify the true causal structure.

Recent advancements in Large Language Models (LLMs) have sparked interest within the causal inference community in exploring whether the world knowledge encoded in LLMs can be leveraged to identify causal graphs [28, 3, 32, 54]. However, many of these studies perform experiments on well-known benchmark datasets such as `BNLearn`, and tend to promote the narrative that standalone LLM-based approaches, which disregard observational data, can significantly outperform traditional data-driven methods. The validity of such claims is questionable if these benchmarks were part of the LLMs' pretraining data. As also pointed out by [55], LLMs may appear to reason causally, but in reality, they could be merely reproducing patterns memorized during training. Extending this direction, Jin et al. propose benchmarks [24, 23] that remove any domain knowledge from the questions posed to LLMs and find that LLMs fail at inferring causal relationships reliably.

While there is an active debate on whether LLMs can genuinely do causal reasoning, the practical relevance of such a capability for modern scientific studies has not received attention. Since one of the main goals of causal graph discovery is to support scientific discovery, in this paper, we focus on the potential of LLM-based graph discovery for science. Notably, we observe that most scientific studies often involve a combination of known variables and novel variables. Even if LLMs are purely memorizing scientific facts, this capability can be useful to provide causal relationships among the known variables and thus accelerate the process of building a graph for a scientist. And in case they have a potential of inferring relationships more generally, that could be even more useful. Realizing this potential, however, requires immediate attention of the research community on the following two positions: **1)** principled evaluation and **2)** progress on new kind of methods that combine LLM-based and data-based approaches (*hybrid* methods).

**P.1. Principled evaluation protocols are needed that prevent dataset leakage**, thereby ensuring that any observed performance gains can be attributed to the genuine causal reasoning capabilities of LLMs rather than inadvertent memorization; and that such gains will translate to performance on real scientific studies.

**P.2. Hybrid methods, that combine LLM-based and data-based approaches, need to be developed to make causal discovery algorithms practical for scientists.** While there is some

initial work in this direction [27, 4, 2, 52, 11], we believe that there is potential to develop more advanced methods that optimally utilize both domain knowledge and data, with the goal of significantly improving accuracy on real-world scientific data.

**Principled, Science-grounded evaluation benchmarks.** In support of **P.1**, we first show that widely used benchmarks in the LLM-based discovery literature, such as those from the BNLearn [43] repository, are memorized by state-of-the-art LLMs. Specifically, we develop a memorization test for causal graphs and find that LLMs such as GPT-4 can complete the information in such benchmarks with near-perfect accuracy, given only a partial peek into the benchmark data.

To avoid memorization concerns, we advocate for *Novel Science-grounded benchmarks* that are based on latest scientific studies focused on causality. If we ensure that all materials of the study were released after the training cut-off date of a LLM, we can rule out *verbatim* memorization of the causal graph studied. Moreover, high accuracy on such a benchmark task mirrors the real-world scientific task, where a scientist may apply a previously trained LLM to a novel scenario of interest and attempt to build a graph. To demonstrate the idea, we collect *four* novel causal graphs curated through expert consensus and sourced from scientific papers published *after* the training cutoff of major LLMs and provide an accompanying codebase for evaluating statistical, LLM-only, and hybrid causal discovery methods. Our dataset collection approach offers a principled and generalizable recipe for constructing robust benchmarks: draw on latest studies to eliminate the risk of verbatim memorization (see Figure 1(a)).

**Hybrid graph discovery methods.** In support of **P.2**, results on the novel benchmarks show that LLM-based methods yield significantly lower performance compared to the results typically reported on the BNLearn datasets. For example, Figure 1(b) shows that a popular LLM-based method, LLM-BFS [25], achieves an F1 score of 0.54 on the Alzheimer's dataset (11 nodes, 19 edges) sourced from a recent study [1], in contrast to 0.93 on the similarly sized Asia graph from BNLearn. Notably, the improvement due to LLM-based methods compared to the existing data-based algorithms such as the PC algorithm is markedly smaller on the Alzheimer's dataset than on Asia, challenging a growing narrative in recent literature that LLM-only methods are often adequate for causal discovery. This insight underscores the need for principled methods that integrate two *complementary* sources of information: (a) world knowledge encoded in LLMs, and (b) statistical signals inferred from data. To encourage progress in this direction, we consider a simple hybrid extension of the PC algorithm that uses the LLM-predicted graph as a prior (shown as the red bar in Figure 1(b)), and show that, despite its simplicity, this hybrid approach outperforms both standalone statistical and LLM-based methods, achieving an F1 score of 0.67 on the novel Alzheimer's graph.

## 2 P.1: Call for Robust Benchmarks

In this section, we critically examine existing causal discovery benchmarks to assess their suitability for benchmarking LLM-based causal discovery performance. Our analysis reveals key limitations, motivating the need for a principled approach to constructing novel benchmarks. We articulate the following insights as part of our first position:

**P.1a** Popular benchmarks, such as those from BNLearn, have been memorized by LLMs thus undermining their utility for benchmarking LLM-based causal discovery.

**P.1b** We must develop novel benchmarks grounded in scientific literature released after LLM's training to mitigate the risk of verbatim memorization and enable a genuine assessment of their causal reasoning capabilities.

### 2.1 P.1a: Current Causal Benchmarks Fall Short for LLM-Based Causal Discovery

Detecting whether an LLM has memorized a given graph benchmark is particularly challenging for closed-source models such as GPT-4 since their training data and mechanisms remain opaque. Prior work [7] has shown that mere presence of data sequences in pretraining corpora does not necessarily mean memorization; rather, memorization depends on factors such as model size and data frequency, thereby undermining the assumption that any pretraining overlap invalidates a benchmark. Hence, successful memorization tests in the literature often use prompting techniques that provide partial datasets and ask LLMs to complete the missing portions. When such prompts lead to exact reproduction, the most plausible explanation is memorization, as reasoning alone cannot justify

| Dataset | M1 | | | | M2 | | | | M3 (Nodes) | | | | M3 (Edges) | | | |
|---|---|---|---|---|---|---|---|---|---|---|---|---|---|---|---|---|
| | 0% | 25% | 50% | 75% | 0% | 25% | 50% | 75% | 0% | 25% | 50% | 75% | 0% | 25% | 50% | 75% |
| Asia | 0.75 | 0.91 | **1.00** | **1.00** | **1.00** | **1.00** | **1.00** | **1.00** | 0.25 | **1.00** | **1.00** | **1.00** | 0.00 | **1.00** | **1.00** | **1.00** |
| Cancer | **1.00** | **1.00** | **1.00** | 0.5 | **1.00** | **1.00** | **1.00** | **1.00** | **1.00** | **1.00** | 0.80 | **1.00** | **1.00** | 0.86 | **1.00** | 0.67 |
| Earthquake | 0.60 | 0.75 | 0.67 | 0.50 | **1.00** | **1.00** | **1.00** | **1.00** | **1.00** | 0.86 | **1.00** | 1.00 | **1.00** | **1.00** | **1.00** | **1.00** |
| Child | **1.00** | 0.11 | 0.06 | 0.53 | 0.44 | 0.55 | 0.44 | 0.36 | **1.00** | 0.85 | 0.89 | 0.00 | 0.17 | 0.00 | 0.30 | 0.16 |
| Insurance | 0.06 | 0.11 | 0.45 | 0.59 | 0.36 | 0.44 | 0.24 | 0.00 | 0.21 | 0.25 | 0.92 | 0.77 | 0.00 | 0.00 | 0.00 | 0.00 |
| Alarm | **1.00** | 0.72 | 0.79 | 0.89 | 0.49 | 0.38 | 0.12 | 0.00 | 0.97 | 0.72 | 0.92 | 0.00 | 0.43 | 0.10 | 0.00 | 0.00 |

Table 1: F1 scores for memorization tests (M1–M3) across datasets at varying context levels ($\alpha$). High F1, especially at low $\alpha$, indicates memorization.

| Causal Graph | Nodes | Edges | Colliders | In-Degree | | | Longest Path |
|---|---|---|---|---|---|---|---|
| | | | | Min | Median | Max | |
| Alzheimer's | 11 | 19 | 1 | 0 | 2 | 4 | 5 |
| COVID-19 Respiratory | 11 | 20 | 1 | 0 | 2 | 4 | 7 |
| Sweden Transport | 11 | 10 | 3 | 0 | 1 | 3 | 3 |
| COVID-19 Complications | 63 | 138 | 23 | 0 | 2 | 7 | 23 |

Table 2: Characteristics of Novel Science Datasets included in our paper.

verbatim recall of real-world data. Reconstruction-based tests of this kind exist for tabular data [6], images [30], text [35, 5, 18, 8, 29], etc. We extend such tests for causal graphs.

Specifically, we perform reconstruction-based memorization tests to evaluate the credibility of widely-used benchmarks in causal graph discovery. We prompt the LLM with partial knowledge about specific aspects of a dataset and ask it to infer or complete the missing components. Since our focus is on causal graphs, we identify three natural and meaningful categories of information against which to assess memorization. These are outlined below.

**M1** Given the dataset name and a random $\alpha\%$ subset of nodes, predict the remaining nodes.

**M2** Provided with the dataset name, the full list of nodes, and an $\alpha\%$ subset of edges in the prompt, identify the remaining nodes.

**M3** Given the dataset name and a subgraph induced by a random $\alpha\%$ subset of nodes (with intra-subset edges), complete the rest of the nodes and edges.

Table 1 presents the results of our memorization tests, with the exact prompts provided in Appendix C. We highlight several key observations:

• Several datasets exhibit near-perfect F1 scores, even at $\alpha = 0\%$, where no contextual information is provided to the LLM. Such performance strongly suggests that the LLM has memorized these datasets, raising concerns about their suitability for evaluation.
• M2 achieves very high F1 scores, even at $\alpha = 0$, demonstrating that LLMs can accurately reconstruct edge structures when provided only with the node list. This raises the question: Do we need sophisticated traversal strategies, such as LLM-BFS, for these datasets.
• LLM performance degrades as graph size increases, as seen in the lower scores for the `Child` and `Insurance` datasets.
• Overall, these results call into question the validity of current benchmarks for evaluating causal reasoning in LLMs and underscore the need for novel, leakage-free benchmarks.

## 2.2 P.1b: Need Science-Grounded Causal Datasets for Benchmarking

The above results suggest that widely-used `BNLearn` graph datasets are likely memorized by LLMs. Therefore, it is important to create novel datasets. Existing literature tackles this problem by creating datasets without any real-world domain knowledge [24, 9] or with synthetic, toy-level scenarios that can be randomized [23]. Although creating a dataset with completely novel causal relationships is useful for evaluating genuine causal reasoning abilities of LLMs, they do not help assess the real-world utility of LLMs for scientific studies. We therefore posit that it is equally important to develop realistic benchmarks for causal discovery that closely mimic the challenges faced by a typical scientific study, as `BNLearn` benchmark did a few decades ago.

In this section, we show how it is possible. We outline a practical recipe for constructing *novel* science-grounded datasets to support robust evaluation of causal discovery algorithms. Our proposal involves: **a)** Finding recent scientific studies that explicitly provide a causal graph (or contain enough

| Dataset | M1 | | | | M2 | | | | M3 (Nodes) | | | | M3 (Edges) | | | |
|---|---|---|---|---|---|---|---|---|---|---|---|---|---|---|---|---|
| | 0% | 25% | 50% | 75% | 0% | 25% | 50% | 75% | 0% | 25% | 50% | 75% | 0% | 25% | 50% | 75% |
| Alz. | 0.00 | 0.11 | 0.12 | 0.00 | 0.00 | 0.00 | 0.00 | 0.00 | 0.00 | 0.62 | 0.67 | 0.80 | 0.00 | 0.34 | 0.23 | 0.00 |
| C19-small | 0.00 | 0.00 | 0.00 | 0.00 | 0.00 | 0.00 | 0.00 | 0.00 | 0.00 | 0.00 | 0.91 | 1.00 | 0.00 | 0.00 | 0.12 | 0.00 |
| C19-large | 0.00 | 0.00 | 0.00 | 0.00 | 0.00 | 0.00 | 0.00 | 0.00 | 0.00 | 0.00 | 0.00 | 0.00 | 0.00 | 0.00 | 0.00 | 0.00 |
| Sweden | 0.00 | 0.00 | 0.00 | 0.00 | 0.00 | 0.00 | 0.00 | 0.00 | 0.00 | 0.00 | 0.67 | 0.67 | 0.00 | 0.00 | 0.00 | 0.06 |

Table 3: Results of memorization tests conducted on the novel science datasets.

information such that a causal graph can be extracted); **(b)** Extracting the relevant source data for the studies (when available) or generating synthetic data with varying distributions. In a way, we propose restarting and adapting the `BNLearn` initiative for LLM evaluation: sourcing new graphs from research papers and associating them with real or synthetic data.

Below we introduce four new causal graphs, each developed in a recent publication through careful expert elicitation and consensus. Key statistics for these graphs are summarized in Table 2. As new LLMs are introduced, the recipe can be repeated to generate more novel datasets.

**Alzheimer's Graph** The first dataset is the Alzheimer's graph from [1], developed with input from five domain experts. It includes two broad categories of variables: clinical phenotypes (e.g., age, sex, education) and radiological features extracted from MRI scans (e.g., brain and ventricular volumes), as illustrated in Fig. 4. The consensus graph was built by retaining only those edges that were agreed upon by at least two of the five experts. As highlighted in Figure 21 of [1], there is substantial disagreement among the individual expert graphs, underscoring the difficulty for automated methods such as LLMs to infer a consensus graph. Although the graph's structure was developed independently, its variables align with a subset of those used in the Alzheimer's Disease Neuroimaging Initiative [40].

**COVID-19 Respiratory Graph** The second graph models the progression of COVID-19 within the respiratory system, as introduced in [34]. It tracks the disease's path from initial viral entry to pulmonary dysfunction and symptomatic manifestations. The graph was developed through iterative elicitation sessions involving 7–12 domain experts and released on medRxiv in February 2022. Figure 3 presents the graph with color-coded nodes corresponding to different stages of infection: viral entry (pink), lung mechanics (yellow), infection-induced complications (orange), and observable symptoms (cyan). Each variable captures a phase in the progression from infection to respiratory distress. The graph was refined through group workshops and follow-ups, followed by independent expert validation to ensure consensus and accuracy.

**COVID-19 Complications Graph** The third dataset extends the respiratory model to include systemic complications resulting from COVID-19, again from [34]. This graph captures how the virus can affect organs beyond the lungs, such as the heart, liver, kidneys, and vascular system. It includes variables like vascular tone, blood clotting, cardiac inflammation, and ischemia, while retaining key pulmonary indicators such as hypoxemia and hypercapnia (see Fig. 3). Constructed using a similar expert elicitation process, this graph focuses on mapping primary pathways that lead to severe complications, including immune overreactions and multi-organ failure. It distinguishes between observable variables used in clinical monitoring and latent variables that reflect complex physiological states. With 63 nodes and 138 edges, this is the most complex of the four graphs and presents a challenging testbed for causal discovery algorithms.

**The Sweden Traffic Dataset** The Sweden traffic dataset was introduced in a recent study [59] aimed at modeling bus delay propagation through a causal graph. Each node corresponds to a variable that influences delays, such as `arrival_delays`, `dwell_time`, and `scheduled_travel_time`. Unlike the previous three studies, a notable feature of this work is that the true graph is not known since it deals with real-world bus traffic data. Instead, the authors provide expert annotations specifying a subset of edge that should definitely exist, and a subset that are forbidden. Thus, the ground-truth contains not only *positive* edges that should be present in the causal graph but also *negative* edges that must be absent. The dataset is inspired by the General Transit Feed Specification (GTFS), a standardized format for public transit schedules and geographic data. As such, benchmarking causal discovery methods on this dataset holds promise for informing real-world applications in transportation systems analysis.

**Memorization Tests** We conduct the same memorization tests on the four science-grounded datasets to assess potential dataset leakage. As shown in Table 3, many F1 scores are consistently low, often close to zero. *While these results do not conclusively rule out memorization, they provide strong*

*evidence that our science-grounded datasets are significantly more likely to support fair and unbiased evaluation of causal discovery methods compared to other widely used benchmarks.*

**Generating Synthetic Observational Data** For datasets where source data is unavailable, we generate synthetic observational data based on expert-designed causal graphs, following the approach used in the `BNLearn` benchmark. We consider two settings: **(a) Linear** and **(b) Non-Linear**, differing in the form of structural equations used for each node. Data is generated in topological order over the graph. Root nodes are sampled as $\mathbf{x}_i \sim \mathcal{N}(0, 1)$. For non-root nodes, we use: $\mathbf{x}_i \sim f_i(\mathrm{Pa}_i) + \epsilon_i$ where $\mathrm{Pa}_i$ denotes the values of the parents of node $i$, and $\epsilon_i \sim \mathcal{N}(0, 1)$ is an exogenous noise term. In the **Linear** setting, $f_i(\mathrm{Pa}_i) = \mathbf{w}^\top \mathrm{Pa}_i$ with weights $\mathbf{w}$ drawn from $\mathcal{U}(0, 2)$ to ensure consistent scaling across graph depths. In the **Non-Linear** setting, $f_i$ is parameterized by a randomly initialized 3-layer MLP with ReLU activations and four neurons per hidden layer: $f_i(\mathrm{Pa}_i) = \mathrm{MLP}(\mathrm{Pa}_i)$ This setup enables flexible modeling of complex non-linear relationships, as in prior work [61].

Studying performance of LLMs on these novel, science-grounded causal graphs can provide a better evaluation of their real world potential and also highlights the limitations, as we show next.

# 3 P.2: Call for Hybrid Methods

Interestingly, one of the reasons that scientific studies provide causal graphs is to evaluate the performance of graph discovery algorithms. For instance, the Swedish Traffic study was focused on applying data-based discovery algorithms to bus transit data. However, multiple studies in medicine [51], climate science [20] and other fields find that data-based graph discovery algorithms are not sufficient for direct application in scientific contexts. This is due to the fundamental limitations of graph discovery with observational data.

Therefore, we believe that combining domain knowledge with data-based methods can be a fruitful way to improve accuracy of graph discovery and make it practical for scientists. This would involve creative methods that combine LLMs' output with principled causal discovery algorithms. Below we provide motivation on why hybrid algorithms may lead to significant gains. **1)** LLM-only methods are not adequate when evaluated on novel benchmarks; **2)** even a simple attempt at hybridizing PC with LLMs yields promising gains. There are a rich set of questions to be explored around quantifying uncertainty in LLMs' graph outputs and integrating themn with different kinds of discovery algorithms.

**P.2a** LLM-only methods exhibit significantly lower accuracy on our novel, science-grounded datasets, highlighting their current limitations.
**P.2b** Advancing hybrid methods offers a promising path forward, as they can effectively combine the strengths of LLMs and statistical inference to improve causal discovery performance.

**Methods.** Before presenting our experimental results, we briefly outline the methods considered: (a) data-driven/statistical methods, which rely solely on observational datasets to infer the causal graph, (b) LLM-based methods, which rely solely on prompt responses, and (c) hybrid methods, which use both LLMs and observational datasets. Among data-driven methods, we consider score-based approaches such as GES [10] and NOTEARS [60], and for constraint-based methods that rely on conditional independence testing, we include PC [49] and FCI [48]. We also ran two variants of LiNGAM: Direct LiNGAM [47] and ICA LiNGAM [46], both of which assume linear relationships among variables with non-Gaussian noise. In our synthetic datasets, the non-Gaussianity assumption is violated in both settings, while the linearity assumption is additionally violated in the non-linear variant. We further evaluate ANM, which assumes additive noise which holds true in our experiments. Then we considered two state of the art LLM-only approaches: (i) **LLM Pairwise** [28], which queries all $\binom{n}{2}$ node pairs, and (ii) **LLM BFS**, which explores the graph in a breadth-first manner using $O(n)$ prompts. Lastly, as a representative for hybrid approaches, we evaluate LLM+PC, a simple way to combine PC with LLM BFS predictions (Sec. 3.2). We defer a detailed description to Appendix B.

## 3.1 P.2a: LLM-Only Methods Fall Short on Novel Science Datasets

Table 4 summarizes the results of evaluating LLM-only methods on the novel science datasets. Our main finding is that accuracy for LLM-only methods is significantly lower than reported numbers on

`BNLearn` datasets [25]. On Sweden Transport and Covid-19 Complications, the F1 is less than 0.3, whereas it is less than 0.6 for the other two datasets, Covid-19 Respiratory and Alzheimers.

- Among the statistical baselines, the LiNGAM variants achieve the best overall performance.
- Within the LLM-only methods, LLM-BFS stands out as the top performer. Interestingly, it constructs the graph using fewer prompts compared to other LLM-based approaches.
- On the COVID-19 Respiratory Complications dataset, the largest and most complex among the benchmarks, LLM-BFS struggles to maintain contextual coherence as it traverses deeper into the graph. Statistical methods also experience performance degradation on this dataset.

## 3.2 P.2b: Hybrid Methods can Potentially Bridge the Gap

The above results show that there is significant potential for hybrid methods to improve accuracy further. For completeness, we describe and evaluate a simple algorithm below.

**Our hybrid algorithm**, LLM+PC, begins by using the LLM-BFS method to generate a prior graph, denoted by $\mathcal{G}_{\text{prior}}$ (though this prior can come from any suitable LLM-based approach). This prior is then used to guide the PC algorithm, which itself operates in two stages: skeleton discovery and edge orientation. During skeleton discovery, the PC algorithm examines each pair of variables $X$ and $Y$, and searches for a conditioning set $S$ such that $X \perp\!\!\!\perp Y \mid S$. If such a set exists, the algorithm removes the undirected edge $X \leftrightarrow Y$ from the graph. Our hybrid method adjusts this process by incorporating the prior knowledge from $\mathcal{G}_{\text{prior}}$: if the prior includes a directed edge $X \rightarrow Y$ or $X \leftarrow Y$, we prevent the PC algorithm from removing the corresponding undirected edge $X \leftrightarrow Y$, even if conditional independence is detected. In the edge orientation phase, we initialize the directions of edges that appear in $\mathcal{G}_{\text{prior}}$ first, and then allow the PC algorithm to determine the orientation of the remaining undirected edges based on its standard orientation rules.

In Table 4, we evaluate two variants of our hybrid algorithm, differing only in the choice of hypothesis tests used within the PC. One variant uses Fisher's Z-test, which is well-suited for detecting linear dependencies, while the other uses Kernel Conditional Independence (KCI) test to capture non-linear relationships. Across all datasets, we observe that at least one variant consistently achieves the highest F1-score, outperforming both LLM-only and statistical baselines. Importantly, neither variant performs significantly worse on any dataset, underscoring the robustness of hybrid methods. In certain cases, the hybrid methods exhibit slightly lower precision as we constrain the PC algorithm to retain prior edges predicted by the LLM, and this can sometimes include false positives. Results are robust to changes in hyperparameters; for details see App. G.

On the bigger and more complicated Covid-19 Complications dataset, while the hybrid methods retain an edge, the performance gains over other approaches are less pronounced. We believe that the results underscore the importance of new algorithms that can combine domain knowledge and data statistics. Below we highlight the research questions that can be explored by the community and provide some explorations using the simple hybrid algorithm above.

**RQ1** What are other promising ways of combining LLMs with data-based methods? For example, how does adding a post-processing phase to a hybrid algorithm, where edges are selectively removed based on additional hypothesis tests from observational data, affect the accuracy?

**RQ2** How much would adding negative edges as priors improve performance?

**RQ3** How do the observed performance trends generalize to open-source LLMs? And how can we develop novel learning strategies for language models that enable them to learn cause and effect from scientific corpora?

**RQ1: Dropping Edges.** In this section, we evaluate a variant of our LLM+PC algorithm that includes a post-processing step to prune extraneous edges using statistical hypothesis tests on the observational dataset. The PC algorithm can sometimes leave some edges unoriented, resulting in cycles. To ensure acyclicity, we first drop a minimal set of edges from the LLM+PC output to obtain a DAG. For each surviving edge, we identify its minimal separator set (witness set), perform a conditional independence test, and record the corresponding $p$-value. We then sort the edges by ascending $p$-value and remove the top $\alpha\%$. Thus, $\alpha = 0\%$ corresponds to the unaltered LLM+PC output, while higher $\alpha$ values yield sparser graphs. Results in Table 5 show that pruning edges generally degrades performance, with a consistent drop in F1 scores across datasets. These results suggest that, at least for the datasets considered, retaining the original LLM+PC output without aggressive post-hoc

| Methods | Covid-19 Resp. | | | Alzheimers | | | Sweden Transport | | | Covid-19 Compl. | | |
|---|---|---|---|---|---|---|---|---|---|---|---|---|
| | Pre | Rec | F1 | Pre | Rec | F1 | Pre | Rec | F1 | Pre | Rec | F1 |
| GES | 0.25 | 0.10 | 0.14 | 0.08 | 0.05 | 0.06 | 0.27 | 0.27 | 0.27 | - | - | |
| PC(Fisherz) | 0.14 | 0.05 | 0.07 | 0.50 | 0.52 | 0.51 | 0.54 | 0.60 | 0.57 | 0.05 | 0.03 | 0.04 |
| PC(KCI) | 0.33 | 0.10 | 0.15 | 0.36 | 0.21 | 0.27 | 0.28 | 0.4 | 0.33 | 0.05 | 0.01 | 0.02 |
| ICA LiNGAM | 0.44 | 0.2 | 0.28 | 0.58 | 0.52 | 0.55 | **0.71** | 0.50 | 0.59 | **0.07** | 0.01 | 0.01 |
| Direct LiNGAM | 0.33 | 0.10 | 0.15 | 0.50 | 0.10 | 0.17 | 0.62 | 0.50 | 0.55 | 0.00 | 0.00 | |
| ANM | 0.44 | 0.20 | 0.28 | 0.30 | 0.15 | 0.20 | 0.22 | 0.2 | 0.21 | 0.04 | 0.04 | 0.04 |
| FCI | 0.30 | 0.15 | 0.20 | 0.42 | 0.26 | 0.32 | 0.50 | 0.3 | 0.38 | 0.02 | 0.03 | 0.03 |
| LLM pairwise | 0.26 | 0.35 | 0.30 | 0.17 | 0.31 | 0.22 | 0.20 | 0.50 | 0.29 | - | - | |
| LLM BFS | **0.90** | 0.45 | 0.60 | **0.69** | 0.47 | 0.56 | 0.25 | 0.4 | 0.31 | 0.06 | 0.04 | 0.05 |
| PC(Fisherz) + LLM | 0.64 | **0.80** | **0.71** | 0.58 | 0.78 | 0.66 | 0.64 | **0.70** | **0.67** | 0.06 | **0.07** | **0.07** |
| PC(KCI) + LLM | **0.90** | 0.45 | 0.60 | 0.64 | **0.84** | **0.73** | 0.50 | 0.50 | 0.50 | **0.07** | 0.05 | 0.06 |

Table 4: Results on Non-Linear Observational Dataset. GES and LLM-pairwise are compute-intensive methods and were not feasible to run for the larger Covid-19 Complications dataset.

| $\alpha\%$ Edges | COVID-19 Resp. | | | Alzheimer's | | | Sweden Transport | | |
|---|---|---|---|---|---|---|---|---|---|
| | P | R | F1 | P | R | F1 | P | R | F1 |
| 0 | **0.64** | **0.80** | **0.71** | 0.58 | **0.78** | **0.66** | 0.63 | **0.70** | **0.67** |
| 5 | **0.64** | 0.70 | 0.67 | 0.58 | 0.74 | 0.65 | 0.60 | 0.60 | 0.60 |
| 10 | 0.62 | 0.65 | 0.63 | 0.57 | 0.68 | 0.62 | **0.66** | 0.60 | 0.63 |
| 25 | 0.55 | 0.50 | 0.52 | 0.53 | 0.53 | 0.53 | 0.62 | 0.50 | 0.55 |
| 50 | 0.42 | 0.25 | 0.31 | **0.61** | 0.42 | 0.50 | 0.4 | 0.2 | 0.27 |

Table 5: Precision, Recall, and F1 Score after removing edges based on p-Value

pruning yields better results. *Future Question: How robust is this result for other constraint-based algorithms beyond PC?*

**RQ2: Incorporating Priors on Missing Edges.** In this experiment, we ask: should the prior provided to the PC algorithm be limited to edges believed to exist in the causal graph? In practice, we may also possess knowledge about edges that should not exist, what we refer to as *negative edges*. These can come from expert annotations or be inferred heuristically from LLM predictions. For example, if an LLM predicts $k$ edges, any subset of the remaining $\binom{n}{2} - k$ pairs can serve as candidates for negative prior. To assess the value of incorporating such information, we evaluate hybrid performance under two settings. In the Sweden Traffic dataset, prior work identifies a set of ground-truth negative edges. For other datasets, we construct a noisy negative prior by randomly sampling edges not predicted by the LLM, acknowledging that these may include false negatives.

We modify our LLM+PC hybrid algorithm to leverage this information during the skeleton discovery phase: any edge included in the negative prior is forcibly removed from the skeleton, regardless of whether the PC algorithm identifies a separating set (i.e., witness set).

- Incorporating ground-truth negative priors enhances performance, as seen in the Sweden Traffic dataset in Tab. 6 (left). Specifically, the PC+LLM (WITH NEGATIVE PRIOR) method shows significant gains in precision without loss in recall, leading to improved F1 scores.
- For datasets where negative priors are derived from LLM outputs, the improvements are less consistent due to potential noise in the inferred prior. Table 6 (right) presents results across varying levels of $\alpha$, where $\alpha$ denotes the percentage of edges absent in the LLM-only prediction that are used as negative priors. Consequently, $\alpha = 0$ corresponds to the standard PC+LLM method, while $\alpha = 100$ reflects LLM-only predictions. We see that our original PC+LLM achieves the best F1 here.
- These results illustrate that although negative priors can be beneficial, their effectiveness is highly sensitive to their quality—noisy or incorrect priors may in fact impair overall performance. More work is needed to fully answer this question and explore choices in both prior and algorithm design.

**RQ3: Extensions using Open-Source LLMs.** Another key question is whether we can move away from propietary LLMs and develop methods using open LLMs. Two research directions are: **1)** How to train language models to infer cause-effect relationships based on a corpus of documents? For instance, can we train specific models for domains such as biomedical or climate science? **2** How do

| | P | R | F1 |
|---|---|---|---|
| PC | 0.54 | 0.60 | 0.57 |
| PC+LLM | 0.64 | **0.70** | 0.67 |
| PC+LLM (-ve prior) | **0.70** | **0.70** | **0.70** |

| $\alpha\%$ | COVID-19 Resp. | | | Alz. | | |
|---|---|---|---|---|---|---|
| | P | R | F1 | P | R | F1 |
| 100 (LLM) | **0.90** | 0.45 | 0.60 | **0.69** | 0.47 | 0.56 |
| 0 (PC+LLM) | 0.57 | **0.70** | **0.63** | 0.56 | **0.70** | **0.62** |
| 25 | 0.60 | 0.62 | 0.61 | 0.56 | 0.61 | 0.58 |
| 50 | 0.60 | 0.54 | 0.50 | 0.55 | 0.55 | 0.55 |
| 75 | 0.67 | 0.48 | 0.56 | 0.61 | 0.49 | 0.54 |

Table 6: Left: Sweden dataset with expert-provided ground-truth negative prior. Right: PC+LLM performance under varying levels of randomly sampled LLM-derived negative priors.

we combine language models with existing discovery methods, such that training and/or inference may happen end-to-end?

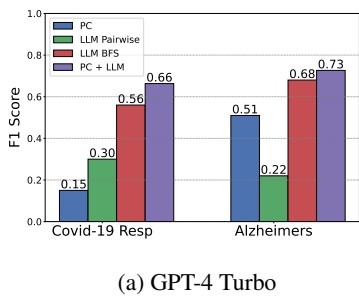

(a) GPT-4 Turbo

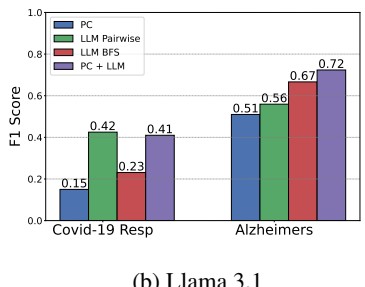

(b) Llama 3.1

Figure 2: Evaluation of GPT-4-Turbo and Llama 3.1 models on NovoGraphs benchmarks. Notably, these models were trained after the release of these datasets, so non-memorization is not guaranteed.

# 4 Conclusion and the Path Forward

We envision a future where causal discovery can be a valuable tool for scientific studies. To support more work in building science-grounded benchmarks and novel algorithms, we aim to open-source all our datasets and code used to constructing the four benchmarks and the hybrid algorithm. Having a general testbed can spur innovation and lead to a robust evaluation of discovery algorithms.

A key concern is whether our benchmark will be relevant as new LLMs keep getting introduced. To test this hypothesis, we also test latest LLMs on the four datasets. we evaluate a recent version of GPT-4 and a recent open language model, `LLaMA 3.1` (2023 checkpoints) on the novel datasets. Both models were released after the publication of novel science datasets. Results are presented in Fig. 2 (nonlinear dataset) and Fig. 7 (linear dataset). In Fig. 2, GPT-4-Turbo yields F1 scores comparable to earlier models, outperforming on the Alzheimer's dataset but slightly underperforming on COVID-19 Resp. for LLM-BFS. These trends are consistent with our broader observations, which is unsurprising because, although the datasets could technically be part of the training corpus, their frequency is likely to be very low which makes memorization unlikely. Notably, LLaMA 3.1 departs from previous patterns: its pairwise comparison strategy surpasses BFS, marking a novel shift in behavior. Across both datasets, our hybrid method consistently outperforms both LLM-BFS and PC alone. These ablations affirm the utility of the benchmark even for newer language models. That said, we would recommend dynamic creation of new benchmarks based on research papers from each future year.

To conclude, we critically examined the limitations of current benchmarking practices in LLM-based causal discovery, highlighting the risks of drawing conclusions without first ruling out dataset leakage. Through a series of targeted memorization experiments, we demonstrated that many widely used benchmarks are vulnerable and often fail to test genuine causal reasoning. To address this gap, we introduced a lightweight yet powerful strategy for building more robust benchmarks grounded in scientific knowledge and human consensus. We hope this sets a new standard for evaluating causal discovery methods and the community builds more science-grounded benchmarks to evaluate progress. Finally, we advocate for deeper investment in hybrid approaches: methods that can harness the complementary strengths of large language models and observational data. As our results suggest, such integration may hold the key to advancing causal discovery as a key part of scientific studies.

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

## A    Code

We release the anonymous code at the url : https://anonymous.4open.science/r/novographs-5005/

## B    Background: Types of causal discovery algorithms

We categorize related work into: (a) *data-driven* methods, which rely solely on observational datasets to infer the causal graph, (b) *LLM-based* methods, which rely solely on prompt responses, and (c) *hybrid* methods, which use both LLMs and observational datasets.

**Data-Driven Methods.** Constraint-based methods for causal discovery, such as the PC algorithm [49] and the FCI algorithm [48], identify causal relationships by testing conditional independencies. Variants of these methods [12, 42, 21, 53, 56, 44] aim to improve scalability and accommodate different assumptions. While some of these methods offer asymptotic consistency guarantees, their performance in practice often depends on the power of the statistical hypothesis tests applied to determine conditional independencies from observational data, a factor we examine in our experiments. Standard tests include Fisher's z-test [50] for linear dependencies and kernel-based tests [58] for non-linear dependencies. Other methods include score-based methods [10, 36, 22, 41] that optimize a score function over graphs, including recent versions based on continuous optimization [60]; and parametric methods that assume parametric assumptions about the functional relationships among nodes in a causal graph, e.g., assuming non-gaussian noise [46, 31].

**Leveraging LLMs for learning Causal Graphs.** There is a growing interest in augmenting observational data with meta-knowledge, aiming for improved causal predictions [1]. Large Language Models (LLMs) offer a promising source of such augmentation, requiring minimal manual effort. For instance, the pairwise approach [28, 54, 33] finds the causal graph using prompts like "Does A cause B?" for each pair of nodes, then coalesces the graph based on the responses. While effective, this method requires $O(n^2)$ prompts for $n$ nodes, making it costly. Alternative approaches [25] reduce prompt complexity by building the graph with a breadth-first search. Another recent approach considers querying LLMs over triplet of variables [52].

**Hybrid Approaches.** ALCM [27] is a recent approach that begins with the PC algorithm and subsequently queries the LLM to validate each edge predicted by the PC. Other methods in this category initiate with a prior LLM-based graph and adjust it using observational data [4, 2] or use LLM as a post-processing critic for data-based output [32, 52]. [11] introduces a method that adaptively defers to either expert (LLM) recommendations or data-driven causal discovery based on their reliability. In their work, [25] presented a variant that incorporates the $p$-values from statistical tests into the prompts while constructing the causal graph. However, the authors found that the inclusion of $p$-values does not yield any improvement over their standalone LLM variant. This indicates that merely adding superficial data statistics to the prompts is less effective, highlighting the necessity for explicit mechanisms to integrate LLM and data-driven graph predictions, and for testing such mechanisms on non-memorized benchmarks.

However, almost all of the above studies use popular, existing graph datasets such as bnlearn for evaluation of LLM-based methods. In the next section, we show why such evaluation is not reliable.

## C    Prompts for Memorization Tasks

---

**Prompt Template for M1 Task**

You are provided with the name of the `bnlearn` dataset: `{dataset_name}` and the following nodes: `{given_nodes}`. Give me the remaining nodes. Strictly output the nodes in the format: `['node1', 'node2', 'node3']`.

**Note:** Add `bnlearn` if it is a `bnlearn` dataset.

---

## D  Visualizations of Novel Sciences benchmark

The Covid-19 respiratory dataset represents the full pathway of Covid-19's impact on the body, organized into six distinct subsystems: vascular, pulmonary, cardiac, system-wide, background, and other organs. This dataset provides a comprehensive view of Covid-19's effects as observed across various aspects of human anatomy.

The complexity of this dataset stems from the high level of interconnections between the subsystems, resulting in a dense causal graph structure with 63 nodes and 138 edges. This density, along with numerous collider structures, makes it exceptionally challenging to analyze, even with advanced statistical algorithms and causal discovery methods.

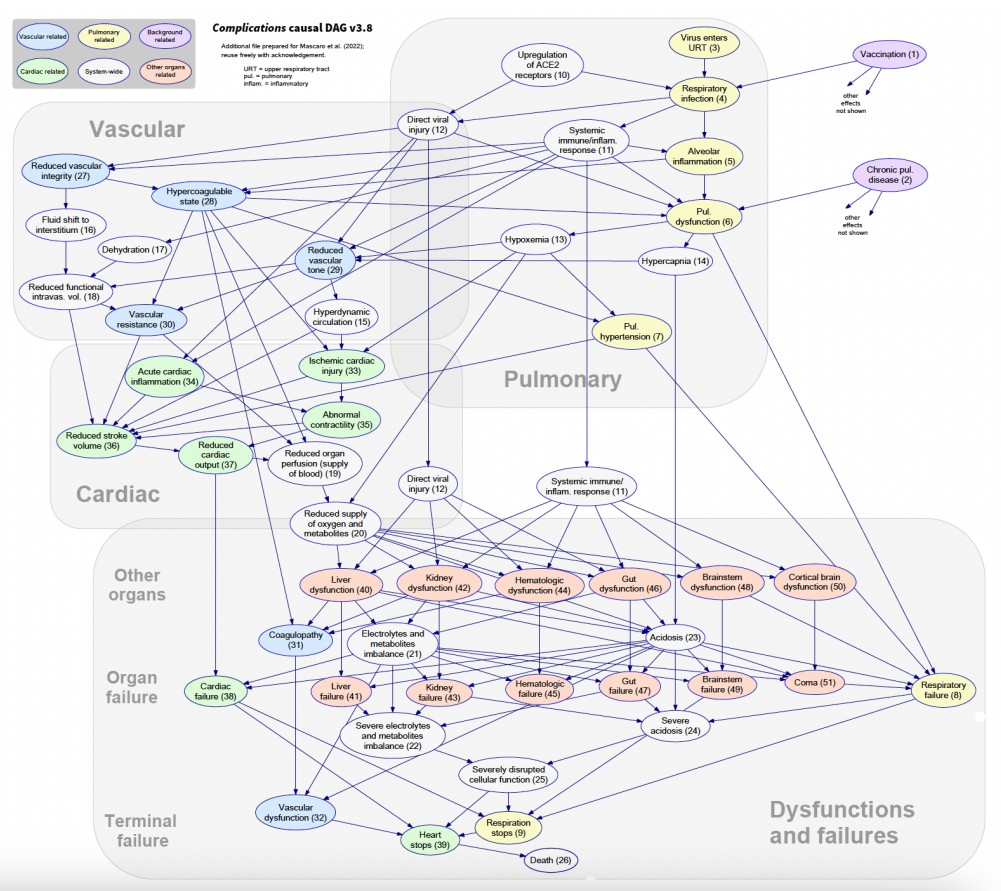

Figure 3: Covid-19 Complications Graph, reproduced from [34].

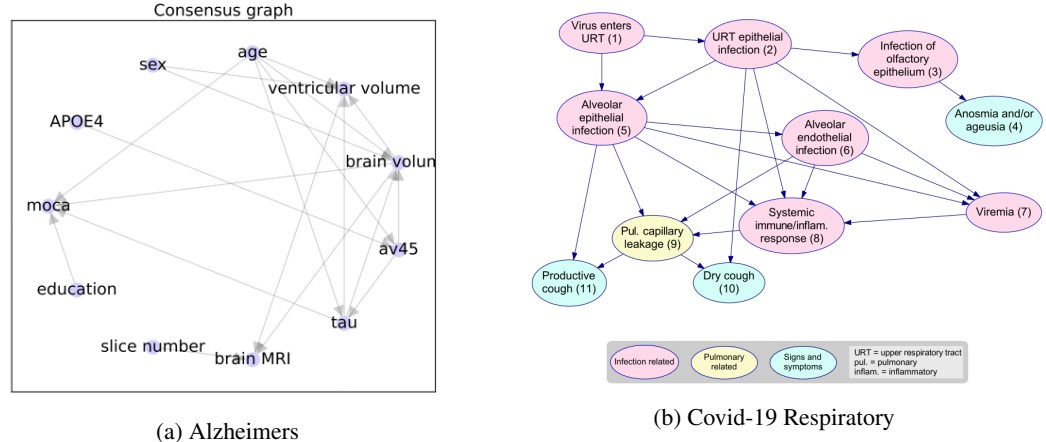

(a) Alzheimers

(b) Covid-19 Respiratory

Figure 4: Consensus causal graphs for Alzheimers benchmark reproduced from [1], and Covid-19 Respiratory dataset reproduced from [34].

## D.1 Sweden Urban Bus Operation Delays (Sweden Transport) Dataset Description

The Sweden Transport dataset [59] contains temporal and operational information from a public bus network. The variables in the dataset are defined as follows:

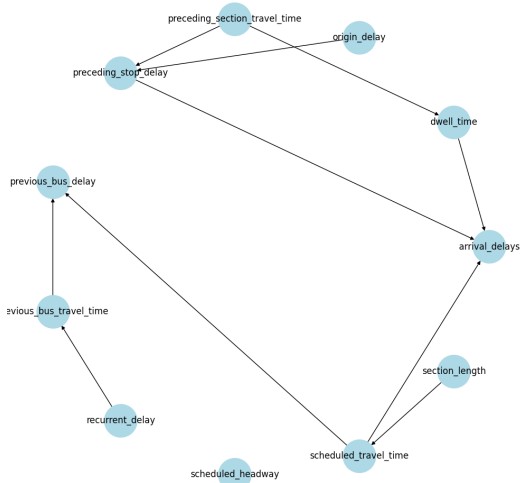

Figure 5: Causal graph obtained from the Sweden Urban Bus Operation Delays dataset.

| True assertion | False assertion |
| --- | --- |
| Preceding stop delay → arrival delay | Dwell time → preceding stop delay |
| Dwell time → arrival delay | Dwell time → preceding section travel time |
| Scheduled travel time → arrival delay | Scheduled headway → dwell time |
| Scheduled travel time → previous bus delay | Section length → preceding section travel time |
| Preceding section travel time → preceding stop delay | Preceding stop delay → preceding section travel time |
| Previous bus travel time → previous bus delay | Previous bus delay → previous bus travel time |
| Recurrent delay → previous bus travel time | Preceding stop delay → previous bus delay |
| Origin delay → preceding stop delay | Scheduled travel time → preceding section travel time |
| Preceding section travel time → dwell time | Section length → origin delay |
| Section length → scheduled travel time | Origin delay → previous bus delay |

Figure 6: Edges obtained from the Sweden Transport dataset. Both positive and negative causal edges are shown. These tables are quoted from the original paper [59] for ease of reference.

- **Arrival Delays**: Arrival delay of bus $j$ at stop $i$; the difference between the actual arrival time and the scheduled arrival time.
- **Dwell Time**: Actual dwell time at the preceding stop $(i-1)$; the difference between actual departure and arrival time at stop $i-1$ for bus $j$.
- **Preceding Section Travel Time**: Actual running time between stops $i-2$ and $i-1$; the difference between arrival at $i-1$ and departure from $i-2$.
- **Scheduled Travel Time**: Scheduled running time between stops $i-1$ and $i$; the difference between scheduled arrival at $i$ and scheduled departure from $i-1$.
- **Preceding Stop Delay**: Arrival delay of bus $j$ at stop $i-1$; the difference between actual and scheduled arrival time at stop $i-1$.
- **Previous Bus Delay**: Arrival delay (knock-on effect) of preceding bus $j-1$ at stop $i$; the difference between its actual and scheduled arrival time.
- **Previous Bus Travel Time**: Actual running time of bus $j-1$ between stops $i-1$ and $i$; used to indicate current traffic conditions.
- **Recurrent Delay**: Historical mean travel time of bus $j$ at stop $i$ during the same hour on weekdays; reflects recurrent congestion patterns.
- **Origin Delay**: Departure delay of bus $j$ at the first stop; the difference between actual and scheduled departure time.
- **Scheduled Headway**: Planned time interval between arrival times of buses $j-1$ and $j$ at stop $i$.
- **Section Length**: Distance between stop $i-1$ and $i$ (in metres).

Table 7: Results on Linear Observational Dataset.

| methods | Covid-19 Resp. | | | Alzheimers | | | Sweden Transport | | | Covid-19 Compl. | | |
|---|---|---|---|---|---|---|---|---|---|---|---|---|
| | Pre | Rec | F1 | Pre | Rec | F1 | Pre | Rec | F1 | Pre | Rec | F1 |
| GES | 0.16 | 0.20 | 0.18 | 0.26 | 0.26 | 0.26 | 0.21 | 0.30 | 0.25 | - | - | |
| PC(Fisherz) | 0.31 | 0.45 | 0.37 | 0.47 | 0.47 | 0.47 | 0.44 | **0.80** | 0.57 | 0.04 | 0.02 | 0.03 |
| PC(KCI) | 0.27 | 0.25 | 0.26 | 0.57 | 0.42 | 0.48 | **0.66** | **0.80** | **0.72** | 0.03 | 0.015 | 0.02 |
| NOTEARS | 0.13 | 0.10 | 0.11 | 0.16 | 0.26 | 0.20 | 0.16 | 0.20 | 0.18 | - | - | |
| ICA LiNGAM | 0.25 | 0.20 | 0.22 | 0.11 | 0.26 | 0.15 | 0.21 | 0.30 | 0.25 | 0.05 | 0.17 | 0.07 |
| Direct LiNGAM | 0.18 | 0.35 | 0.24 | 0.20 | 0.30 | 0.24 | 0.16 | 0.30 | 0.21 | 0.03 | 0.17 | 0.05 |
| ANM | 0.25 | 0.20 | 0.22 | 0.19 | 0.2 | 0.19 | 0 | 0 | - | 0.04 | **0.58** | **0.07** |
| FCI | 0.12 | 0.15 | 0.13 | 0.60 | 0.16 | 0.25 | 0.50 | 0.40 | 0.44 | 0.04 | 0.01 | 0.01 |
| LLM Pairwise | 0.26 | 0.35 | 0.30 | 0.17 | 0.31 | 0.22 | 0.20 | 0.50 | 0.29 | - | - | |
| LLM BFS | **0.90** | 0.45 | 0.60 | **0.69** | 0.47 | 0.56 | 0.25 | 0.40 | 0.31 | **0.06** | 0.04 | 0.05 |
| PC(Fisherz) + LLM | 0.46 | **0.70** | 0.56 | 0.54 | 0.68 | 0.60 | 0.53 | **0.80** | 0.64 | **0.06** | 0.06 | 0.06 |
| PC(KCI) + LLM | 0.63 | 0.60 | **0.61** | 0.60 | **0.78** | **0.68** | **0.66** | **0.80** | **0.72** | **0.06** | 0.05 | 0.05 |

## E    Results on Linear Observational Dataset

Statistical methods were applied to linearly generated data, and results were obtained using GPT-4 with a 2021 cutoff, facilitating a comparison of performance between traditional algorithms, the LLM-based approach and our hybrid method.

## F    Ablations using Linear dataset

We conduct ablation studies using GPT-4 Turbo and LLaMA 3.1 on linearly generated data and observed that our hybrid PC+LLM method outperforms both individual baselines. This demonstrates the advantage of combining PC's statistical rigor with LLM's contextual reasoning for causal discovery.

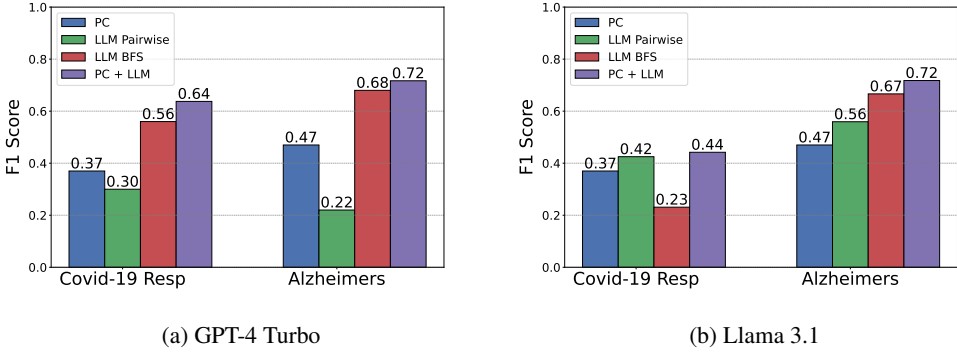

(a) GPT-4 Turbo

(b) Llama 3.1

Figure 7: Evaluation of GPT-4-Turbo and Llama 3.1 models on Novel Sciences benchmarks. Notably, these models were trained after the release of these datasets, so there is a possibility that they may have encountered our datasets during training.

## G    Experiments for Research Question: RQ4

We conduct a series of ablation studies to assess the robustness and generalization ability of our hybrid PC+LLM approach under various modifications to the data generation process.

**Ablation 1: MLP Depth.** We evaluate the impact of increasing the depth of the nonlinear generators by replacing 3-layer MLPs in our default setting with 5-layer MLPs. The results in Table 8 (left) indicate that performance remains consistent, suggesting insensitivity to architectural depth.

627 **Ablation 2: Noise Distribution.** To assess robustness under different exogenous noise assumptions,
628 we replace the default $\mathcal{N}(0,1)$ noise with $\mathcal{N}(0,0.1)$ and $\mathcal{U}(0,1)$. As shown in Table 8 (right),
629 PC+LLM consistently outperforms the PC baseline across all settings.

| Method | COVID-19 Resp. | | | Alzheimers | | |
|---|---|---|---|---|---|---|
| | P | R | F1 | P | R | F1 |
| LLM | **0.90** | 0.45 | 0.60 | **0.69** | 0.47 | 0.56 |
| PC | 0.35 | 0.25 | 0.30 | 0.44 | 0.42 | 0.43 |
| PC + LLM | 0.73 | **0.55** | **0.63** | 0.60 | **0.78** | **0.68** |

| Noise | Method | COVID-19 Resp. | | | Alzheimers | | |
|---|---|---|---|---|---|---|---|
| | | P | R | F1 | P | R | F1 |
| $\mathcal{N}(0,0.1)$ | PC | 0.34 | 0.40 | 0.37 | 0.38 | 0.37 | 0.38 |
| | PC+LLM | **0.58** | **0.70** | **0.63** | **0.56** | **0.68** | **0.62** |
| $\mathcal{U}(0,1)$ | PC | 0.60 | 0.30 | 0.40 | 0.58 | 0.36 | 0.45 |
| | PC+LLM | **0.85** | **0.60** | **0.70** | **0.60** | **0.63** | **0.62** |

Table 8: Left: Effect of deeper MLPs on performance. Right: Performance under noisy LLM-derived priors.

630 **Ablation 3: MLP Initialization.** We compare three initialization strategies for MLP weights:
631 uniform $\mathcal{U}(0,1)$, standard normal, and Xavier normal. As seen in Table 9 (left), the hybrid method
632 retains its advantage across all configurations.

633 **Ablation 4: Linear Coefficient Sampling.** We vary the distribution used for sampling linear SEM
634 coefficients, testing $\mathcal{U}(0,2)$, $\mathcal{N}(0,2)$, and $\mathcal{U}(-1,1)$. Table 9 (right) shows that PC+LLM consistently
635 achieves superior recall and F1 scores.

| Init. | Method | COVID-19 Resp. | | | Alzheimers | | |
|---|---|---|---|---|---|---|---|
| | | P | R | F1 | P | R | F1 |
| Std Normal | PC | 0.44 | 0.20 | 0.28 | 0.35 | 0.37 | 0.36 |
| | PC+LLM | **0.73** | **0.55** | **0.63** | **0.50** | **0.57** | **0.54** |
| Xavier Normal | PC | 0.38 | 0.40 | 0.39 | 0.40 | 0.47 | 0.43 |
| | PC+LLM | **0.59** | **0.80** | **0.68** | **0.54** | **0.68** | **0.60** |

| Coeff. Dist. | Method | COVID-19 Resp. | | | Alzheimers | | |
|---|---|---|---|---|---|---|---|
| | | P | R | F1 | P | R | F1 |
| $\mathcal{N}(0,2)$ | PC | 0.14 | 0.20 | 0.16 | 0.44 | 0.42 | 0.43 |
| | PC+LLM | **0.66** | **0.70** | **0.68** | **0.59** | **0.68** | **0.63** |
| $\mathcal{U}(-1,1)$ | PC | 0.26 | 0.55 | 0.36 | 0.48 | 0.63 | 0.54 |
| | PC+LLM | **0.59** | **0.65** | **0.62** | **0.53** | **0.79** | **0.64** |

Table 9: Left: Performance across different MLP initializations. Right: Effect of different coefficient sampling distributions.

636 **In Summary,** these results collectively demonstrate the robustness and effectiveness of our method
637 across a wide range of data-generating assumptions. Across all ablations, our PC+LLM hybrid
638 approach consistently outperforms the standalone PC method. These experiments effectively illustrate
639 the robustness of hybrid approaches.

