# OpenReview forum: "Realizing LLMs’ Causal Potential Requires Science-Grounded, Novel Benchmarks"
_NeurIPS.cc/2025/Position_Paper_Track — Submitted to NeurIPS 2025 Position Paper Track_

### Official Review · Reviewer_Nj9D · 2025-08-09

**Significance:** 4
**Presentation:** 3
**Rating:** 7
**Confidence:** 4

**Summary:**

The paper challenges the recent success of LLMs for causal discovery. It poses two positions:
P.1. Evaluation protocals are needed which are not diluted by memorization of LLMs.
P.2. LLMs need to be combined with observational data to support causal discovery.

**Strengths:**

- The paper is well structured and written.
- The ideas can be easily followed.
- Paper takes up an very important point for the AI community.
- Data example is very illustrative.

**Weaknesses:**

- No alternative views are mentioned.

**Questions:**

- The paper covers causal discovery. How about causal inference in general?

**Alternative Position:**

No

**Author Identification:**

No.

**Context:**

3

**Discussion:**

4

**Ethics:**

["NO or VERY MINOR ethics concerns only"]

**Position:**

Yes, the paper argues for or against a position related to machine learning.

**Support:**

4

**Thoroughness:**

3

---

### Official Review · Reviewer_EBvG · 2025-08-12

**Significance:** 4
**Presentation:** 3
**Rating:** 7
**Confidence:** 4

**Summary:**

The authors discuss whether the strong performance by LLMs on causal discovery can be attributed to reasoning capabilities or memorization and question the validity of some of the current benchmarks assessing LLMs in this regard. In particular, they repeatedly demonstrate that performance can be tied to memorization and propose an approach to test against memorization. Furthermore, they consider that LLMs have inherent limitations when performing causal discovery. They suggest overcoming them by complementing LLMs with data-based approaches.

**Strengths:**

(S1) - We consider the ideas to be clearly written and articulated

(S2) - The authors formulate a problem (current benchmarks testing how good LLMs are at extracting causal graphs from text do not test for LLM memorization) and provide experiment results to support their claims. Furthermore, they propose a solution for which they provide the corresponding experimental results (testing on causal graphs that can be formed on data created after the LLM release), showing the solution provides a viable path forward on assessing LLMs' causal potential.

(S3) - The authors recognize inherent limitations of the LLMs when dealing with causal discovery and propose complementing them with statistical signals inferred from data.

**Weaknesses:**

(W1) - The authors draft a procedure to elucidate a causal graph from recently published data, curating it with experts' input. Nevertheless, the authors provide little detail on how the causal graph is created and assessed when compared to the causal graph produced by the LLMs. This weakens the reported results.

(W2) - The authors consider that hybrid methods could bridge the gap, overcoming some notable limitations current LLMs exhibit. Nevertheless, this requires satisfying certain preconditions that were not addressed. We pose some questions on this below.

Minor comments:

(W3) - Figures 1 and 2: make sure the colours are friendly to color-blind individuals.

(W4) - Table 3: Please indicate the metric that is being reported. From the previous tables, we understand this is the F1 metric. Nevertheless, the table caption should help the table information to be self-contained.

**Questions:**

(Q1) (a) How is bibliography retrieved, and how are limits established on it to guarantee the quality of sources and avoid spreading too widely? (b) How do the authors guarantee the scalability of the causal graph generation process? (c) How are the concepts from the causal graph normalized? Do the authors suggest matching or defining some ontology?, (d) How are the causal graphs extracted by the experts compared to the causal graphs generated by LLMs? Do we consider the same normalized concepts as an input? By providing such input to the LLM, do we force some bias/transfer some knowledge? If concepts are not normalized, how do we ensure different kinds of causal graphs extracted from LLMs are fairly compared against each other?

(Q2) (a) what data is required for data-based methods and how can be obtained for the problems at hand?, (b) how much of the causal graph the LLM extracts can be potentially covered/recreated considering the data-based methods?, (c) what is the amount of data required to have a certain level of confidence on the causal relationships extracted from data?, (d) are there some confounders or other factors that could affect the outcomes we see from data-based causality extraction methods?

**Alternative Position:**

Yes, and alternative positions are well-considered and addressed by the argument

**Author Identification:**

No.

**Context:**

3

**Discussion:**

4

**Ethics:**

["NO or VERY MINOR ethics concerns only"]

**Position:**

Yes, the paper argues for or against a position related to machine learning.

**Support:**

3

**Thoroughness:**

5

---

### Official Review · Reviewer_gheP · 2025-08-13

**Significance:** 3
**Presentation:** 3
**Rating:** 7
**Confidence:** 4

**Summary:**

The author claims that P1. Recent leverage of LLM in causal discovery is overestimated, as their training corpus may already include domain knowledge used for conventional causal discovery benchmarks. The author expresses this as data leakage, which hinders fair comparison with LLM and conventional causal discovery methods, thereby LLMs can relatively easily outperform causal discovery methods purely based on observational data. To tackle this, the authors suggest evaluating recent scientific data, which may be leakage-free.  For this, the authors suggest a recipe for extracting a causal graph from recent scientific publications. In those extracted causal graphs, LLMs suffer from a significant performance drop with respect to near-perfect performance in the conventional benchmark. Then, to mitigate the gap of LLM behaviour between benchmarks before or after the training cutoff date, and make use of LLM’s world knowledge for causal discovery, the authors claim P2. It is necessary to design hybrid methods that combine Large Language Models (LLMs) with data-driven statistical approaches.

**Strengths:**

- The suggested Positions 1 and 2 are timely and significant in that recently, many LLM research projects have targeted causal discovery benchmarking. The Positions could effectively promote discussion on how to leverage LLM in causal discovery, involving both fields of causal discovery and LLM reasoning.
- Regarding P1, generating a benchmark dataset following the proposed methods seems valid, and an experiment on this could be effective evidence.
- They proposed a novel memorization test for causal discovery and effectively used this as evidence for their P1. Additionally, this test method provides an effective benchmark for future work, which means the discussion invoked by this paper in this direction.

**Weaknesses:**

- The proposed benchmark causal graph construction method seems to largely depend on a domain expert elicitation process for causal graph notation, which limits future work for following P1 on subsequent datasets.
- Lack of justification to use LLM for causal discovery on a cutting-edge benchmark. Though the author demonstrated that LLMs' prior knowledge is weak on the cutting-edge benchmark after their cut-off, they still suggest integrating LLMs into a unified causal discovery framework. It seems to contradict the experimental evidence the author suggests.

**Questions:**

- If the LLM world knowledge itself is not good, for example, on a cutting-edge dataset, how can a hybrid method improve causal discovery?
- Obtaining causal graph annotations from experts is a widely used method; however, it may be vulnerable to reproduction issues since different sets of experts can provide different causal graph annotations. This problem can be mitigated by basing causal graph annotations on consensus reached by the domain community over a sufficient period of time. However, for the cutting-edge datasets presented in this paper, such consensus is difficult to expect. In this respect, how can we obtain robust and widely agreed-upon causal graphs?

**Alternative Position:**

Yes, and alternative positions are trivial straw-man arguments

**Author Identification:**

No.

**Context:**

3

**Discussion:**

3

**Ethics:**

["NO or VERY MINOR ethics concerns only"]

**Position:**

Yes, the paper argues for or against a position related to machine learning.

**Support:**

2

**Thoroughness:**

3

---

### Note · Authors · 2025-08-26

**1-11 Submit Again:**

Definitely yes

**1-1 Submission Process:**

5

**1-2 Next Year:**

It would be nice to provide the authors with a chance for rebuttal. Sometimes, the positions can be contested, and consensus can be achieved only via discussion.

**1-3 Future Development:**

Adding a rebuttal phase.

**1-4 Interest:**

["Structured debates on controversial topics"]

**1-5 Thoughtful:**

8

**1-6 Supportive:**

10

**1-7 Technical Aspects Versus Position:**

9

**1-8 Gate Keeping:**

10

**1-9 Camera Ready Changes:**

1. We will emphasize that the causal graphs that we considered in the paper are extracted verbatim from published papers and not constructed by us. Each paper proposes a causal graph for their problem of interest.
We will also expand on the dataset construction details further in the appendix, specifically how they were constructed with a consensus over multiple domain experts.

2. We will make the Figure captions more accessible and enhance the readability.

**3-1 Review Response1:**

gheP

**3-2 Reaction To Review1:**

We sincerely thank the reviewer for their thorough and constructive feedback on our work. We are pleased that the reviewer values our proposed approach for cheaply obtaining ground-truth causal graphs, our carefully designed memorization tests, and our experiments.

We agree that eliciting human consensus is inherently difficult and sometimes noisy, and we acknowledge that this remains a broader challenge in causal discovery. Beyond relying on a mixture of domain experts and resolving disagreements through iterative refinement, we are not aware of substantially more robust alternatives. Importantly, in the papers we built upon, the respective authors made deliberate efforts to refine precisely those edges where experts disagreed. Through discussions, the experts reached consensus, which provides us with reasonable confidence in the extracted graphs. However, we agree that ensuring complete robustness is difficult.

On these novel datasets, we concur that LLMs alone can only provide a weak prior (but this information may still be complementary to statistical discovery algorithms). That is why, in our experiments, hybrid methods consistently outperformed both state-of-the-art LLM-only methods and purely statistical causal discovery methods. Given the weak prior, we also conducted several ablations of our PC+LLM approach where we do not provide all the edges returned by LLM, such as dropping prior edges and incorporating priors on negative edges. For the four datasets considered, these variants did not substantially outperform our base PC+LLM configuration.

**3-3 Review Response2:**

EBvG

**3-4 Reaction To Review2:**

We thank the reviewer for a very thorough and constructive review of our work. We are pleased that the reviewer appreciated the clarity of our writing and the formulation of our problem statement.
We would like to emphasize that the causal graphs used in our paper were not created by us but were used verbatim from recent publications. We view this as a strength of our approach: researchers seeking to benchmark LLM-based causal discovery can readily curate reliable benchmarks by drawing from existing peer-reviewed literature. While this strategy cannot entirely eliminate noise, restricting to published and reviewed sources does help reduce it to a reasonable extent.
We acknowledge the lack of detail regarding some of the curated datasets in the main paper. While we included information in the appendix along with inline citations to the scientific papers that provide full descriptions, we will expand the appendix further and include the key details in the main paper.
For improved readability, we will adjust the colors in the figures and explicitly indicate in each table that the reported metric is the F1 score.

**3-5 Review Response3:**

Nj9D

**3-6 Reaction To Review3:**

We sincerely thank the reviewer for their positive feedback on several aspects of our paper. We are glad that our contributions were well received and that there are no major concerns requiring immediate clarification.
While downstream tasks such as causal inference depend on obtaining the correct graph (which we study), we acknowledge that extending our ideas to causal inference is non-trivial and will require substantial additional thought. We defer this as future work.

---

### Meta-Review · Area_Chair_KiqT · 2025-09-14

**Rating:** 7
**Confidence:** 5

**Strengths:**

Reviewers unanimously agree this is timely research given the importance of causal discovery in LLMs. The paper identifies a gap in current benchmarks, proposes a novel memorization test with supporting experimental results, and acknowledges the inherent limitations of LLMs in this context.

**Weaknesses:**

There are concerns about the proposed solution’s dependence on human experts, which limits its applicability, as well as the lack of discussion on alternative positions.

**Questions:**

N/A

**Ethics:**

NO or VERY MINOR ethics concerns only

**Thoroughness:**

5

---

### Decision · Program_Chairs · 2025-09-26

Reject